# Ammonia Concentration in the Eluent Influences Fragmentation Pattern of Triacylglycerols in Mass Spectrometry Analysis

**DOI:** 10.3390/metabo12050452

**Published:** 2022-05-18

**Authors:** Marta Velasco, David Balgoma, Olimpio Montero

**Affiliations:** 1Delegación Institucional Castilla y León, Consejo Superior de Investigaciones Científicas (CSIC), 47003 Valladolid, Spain; m.velasco@dicyl.csic.es; 2Unidad de Excelencia, Instituto de Biología y Genética Molecular (IBGM), Universidad de Valladolid—Consejo Superior de Investigaciones Científicas (CSIC), 47003 Valladolid, Spain; david.balgoma@uva.es; 3Analytical Pharmaceutical Chemistry, Department of Medicinal Chemistry (ILK), Uppsala University, 75123 Uppsala, Sweden

**Keywords:** triacylglycerols, mass spectrometry, regioisomery

## Abstract

Correct assessment of the fatty acyl at the glycerol *sn*-2 position in triacylglycerol (TAG) analysis by liquid chromatography and mass spectrometry (LC-MS) is challenging. Ammonium hydroxide (NH_4_OH) is the preferred choice for the solvent additive for the formation of the ammonium adduct ([M + NH_4_]^+^). In this study, the influence of different NH_4_OH concentrations in the eluents on TAG adduct formation and fragmentation under LC-MS analysis was assessed. Increasing NH_4_OH concentrations delayed the chromatographic elution time according to a power function. The [M + NH_4_]^+^ and [M + ACN + NH_4_]^+^ adducts (where ACN means acetonitrile) were formed at all ammonium concentrations assayed. [M + ACN + NH_4_]^+^ predominated above 18.26 mM [NH_4_OH], and the intensity of [M + NH_4_]^+^ dropped. TAG fragmentation for fatty acyl release in the MS^E^ was reduced with increasing [M + ACN + NH_4_]^+^ adduct, which suggests that ACN stabilizes the adduct in a way that inhibits the rupture of the ester bonds in TAGs. A linear equation (H_sn-I_ = a × H_[M+NH4]+_, where *sn*-I refers to the *sn* position of the glycerol (I = 1, 2, or 3) and H is the peak height) was deduced to quantify the dehydroxydiacylglycerol fragment intensity in relation to [M + NH_4_]^+^ intensity in the full scan. This equation had a slope mean value of 0.369 ± 0.058 for the *sn*-1 and *sn*-3 positions, and of 0.188 ± 0.007 for the *sn*-2 position.

## 1. Introduction

Triacylglycerols (TAGs) have a chemical structure in which three fatty acyl chains esterify to the hydroxyl groups of a glycerol molecule. Depending on whether the three acyl chains are the same or not, and the position of the glycerol backbone they esterify (*sn*-1, *sn*-2, or *sn*-3), different chemical structures arise. These structures can be annotated as XXX when the three acyl chains are the same; XYX or XXY when one of the acyl chains is different; and XYZ, XZY, or YXZ when the three acyl chains are different. They are known as regioisomers. Which acyl chain is placed in the *sn*-2 position has been shown to be relevant in relation to the metabolic fate of TAGs during digestion [1,2,3,4,5].

Research regarding the development of methods for accurate assignation of the acyl chain position is ongoing at present. The present implementation of mass spectrometry (MS) functionalities has brought the opportunity to develop reliable platforms for TAG analysis [5,6]. Presently, MS-based methodologies allow resolving the acyl chain composition for isobaric species as well as the regiospecificity, though accurate quantification remains a challenge. Chain length and number of double bonds account for additional complexity of TAGs. Currently, TAG analysis is conducted in MS under positive ionization with the formation of the ammonium adduct ([M + NH_4_]^+^) with electrospray ionization (ESI), which is submitted to further fragmentation (MS^2^ and MS^3^) in order to characterize the acyl chains. Formation of the protonated ion [M + H]^+^ has been the classical method for longer, but formation of a number of adducts and source fragmentation were shown to be a drawback [7,8,9,10]. The position of the acyl chains in the glycerol backbone may be determined from the relative intensity of the diacylglycerol fragments formed by neutral loss of the esterifying fatty acid as the corresponding aldehyde [11,12,13]. Recently, approaches using sodium (Na^+^) and lithium (Li^+^) TAG adducts have been shown to render better performance of the fragmentation spectrum for quantification [14,15], but ammonium is still the favorite in many laboratories. Additionally, methodologies encompassing shot-gun or direct infusion versus separation by liquid chromatography before MS detection are currently used and are a matter of debate regarding their efficiency [5,6,16]. Therefore, active research concerning the influence of other factors on the fragmentation information quality, as well as the application of diverse methodologies, is ongoing at present regarding TAG analysis [17,18,19,20,21,22,23,24,25].

Since formate or acetate ammonium salts are the most popular eluent additives in LC-MS, but with formic or acetic acid as well as an additive, this study deals with the influence of the ammonium concentration on the fragmentation efficiency. This study uses ammonium hydroxide instead of the salt to avoid interference of other ions, even negative ones, which might negatively influence the equilibrium of the ammonium adduct. In addition, this study takes a chance on the MS^E^ methodology to reduce the analysis to one run.

## 2. Results

### 2.1. Chromatographic Separation of Standards

Diverse elution gradients were checked, with run times ranging from 18 to 24 min. Finally, that shown in the Materials and Methods section was chosen as the optimal one for the subject of this study, even though TAGs with more than 60 carbon atoms and fewer unsaturations did not elute within the run time. A typical chromatogram obtained for the standards with the reported method is shown for 54.78 mM NH_4_OH in Figure 1. For this chromatographic platform, the elution time increased with the ammonium concentration (mM) (Figure 2) according to a power function whose power rose in turn with the chain length, also according to a power function (Y = 4 × 10^−5^ × X^1.97^) (data not shown).

As a general pattern, the ammonium concentration [NH_4_OH] increased the retention times in a minimal way (Figure 2). For an ammonium concentration of 9.13 mM in the mobile phases (0.1% in volume), the effect of the number of carbons and unsaturations of the TAGs depended on the elution range: (i) two additional carbons in the acyl chains rose the elution time to between 0.7 and 3.7 min; and (ii) one double bond decreased the elution time to between 1.5 to 2.9 min (Figure 3). The elution time of TAGs fitted to an exponential equation where the exponent increases with the number of carbons. Conversely, the response in the elution time fitted to a power equation on the number of double bonds (Figure 3). Nonetheless, these results may vary with the column type and other chromatographic factors.

### 2.2. Mass Spectrometry and MS^E^ Fragmentation Patterns

Currently, a concentration of ammonium formate or acetate 10 mM is used as an additive in the eluent for the formation of the [M + NH_4_]^+^ adduct, where M represents the nominal mass of the TAG molecule. Consequently, we studied the influence of the NH_4_^+^ concentration on the fragmentation pattern of TAGs in relation to their fatty acid composition. 

Using the elution gradient indicated in Materials and Methods, two adducts were detected in the mass spectrometer, namely [M + NH_4_]^+^ and [M + ACN + NH_4_]^+^ (where ACN means acetonitrile). [M + 2ACN + NH_4_]^+^ was only detectable above 54.78 mM of NH_4_OH, though with a very low intensity. Similarly, [M + Na]^+^ was detected with a very low intensity. The highest ammonium hydroxide concentration the highest [M + ACN + NH_4_]^+^ adduct formation until achieving a plateau at 76.95 ± 11.67 mM of NH_4_OH was observed. Conversely, the signal of [M + NH_4_]^+^ increased with [NH_4_OH] until 18.26 mM and decreased beyond this point (Figure 4). 

In the MS^E^ fragmentation spectrum (function 2 of Waters’ raw files), the [M + Na]^+^ adduct was detected with an increased intensity compared to that in the full scan spectrum, and with a pattern of variation which paralleled that of the [M + NH_4_]^+^ adduct in function 1 (full scan MS). Conversely, decreased intensity of the [M + Na]^+^ adduct was observed with increasing formation of the [M + ACN + NH_4_]^+^ adduct, that is, with increasing [NH_4_OH]^+^ (data not shown).

Regarding acyl chains, their neutral loss yielded the carbocation of the dehydroxydiacylgliceride (C_n_H_m_O_4_^+^). The intensity of the dehydroxydiacylglycerol resulting from the loss of any acyl chain in the MS^E^ spectrum decreased with increasing [NH_4_OH] (Figure 5). It increased linearly with the signal (peak height or area) of [M + NH_4_]^+^ in the full scan with a positive coefficient, but it decreased quadratically with the signal (peak height or area) of [M + ACN + NH_4_]^+^. The *sn* position of the fatty acid did not affect the trends (Figure 6).

According to the results shown above, it may be deduced that acyl release is reduced with increasing ammonium concentration. For TAG(14:0/18:1/22:0) (the one with three different fatty acids), a close intensity was observed for the loss of the acyl chain in the *sn*-1 and *sn*-3 positions (external). Both C14:0 and C22:0 are saturated, but a slightly higher intensity was indicated for the neutral loss of C22:0 (Figure 6b). Values of the coefficients in the linear equation resulting from the plot of dihydroxydiacylglycerol peak height versus the full scan peak height of the [M + NH_4_]^+^ adduct were averaged for the diverse acyl chain lengths. As a result, the following equations were obtained: H*_sn_*_1/*sn*3_ = 0.369 (±0.058) × H_NH4_ for the *sn*-1 and the *sn*-3 position, and H*_sn_*_2_ = 0.188 (±0.007) × H_NH4_ for the *sn*-2 position. In these equations: (i) H*_sn_*_-1/*sn*-2/*sn*-3_ represents the peak height of the neutral loss of the acyl chains in the MS^E^ mass spectrum from the *sn*-1, 2, or 3 positions, and (ii) H_NH4_ represents the peak height of the [M + NH_4_]^+^ adduct in the full scan mass spectrum. As could be expected, the signal of the neutral loss of the acyl chains from [M + NH_4_]^+^ dropped with the increase of the intensity of [M + ACN + NH_4_]^+^ when [NH_4_OH] was increased (Figure 6). The drop fitted to a quadratic equation.

When using the MS^E^ method, accurate adscription of the acyl chains to each species is difficult when two or three TAGs coelute or are not baseline resolved in the chromatogram. Nonetheless, different molecular characteristics allow the assignation, for example, the relative intensity of fragments to the intensity of the TAG intact molecule adduct in the full scan. In addition, the sum of carbons and unsaturations of the acyl neutral losses should equal the sum of carbons and unsaturations of the parent TAG. Additionally, total intensity of a given fragment will be the accumulation of intensities from co-eluting TAGs. Indeed, identifying two of the three acyl chains is enough to know the full molecule composition. Here, we show the results for a single example. In Figure 7, the MS^E^ mass spectrum for two co-eluting TAGs is shown.

As shown in Table 1, fragments with *m*/*z* 599.5 and 603.5 come from TAG(54:4) and fragments with *m*/*z* 549.5 and 603.5 come from TAG(50:2). The fragment with *m*/*z* 599.5 is rendered after release of a C18:0 acyl chain, which may only be at the *sn*-2 position since the calculated intensity for the *sn*-1/*sn*-3 position is much higher than the measured intensity. For the fragment with *m*/*z* 603.5, the intensity rendered by the C18:1 acyl chains in the *sn*-1/*sn*-3 position would account for a peak intensity of 11,660.4, and then the contribution of the C14:0 should be low and from the *sn*-2 position. Accordingly, the most probable TAG compositions are (18:2/18:0/18:2) and (18:1/14:0/18:1) for TAG(54:4) and TAG(50:2), respectively.

### 2.3. TAGs from Ovine Plasma

TAGs from ovine plasma were extracted and measured at two [NH_4_OH]: 9.13 mM (lower panel in Figure 8) and 54.78 mM (upper panel in Figure 8). Their base peak chromatograms are depicted in Figure 8. About 53 TAGs were identified (Table 2). Elution times were increased in the 54.78 mM [NH_4_OH] compared with the 9.13 mM [NH_4_OH]. Lysophosphatidylcholines (LPCs) and sphingomyelins (SMs) eluted at the early times, while TAGs started to elute at about 1.7 min (Table 2 and Appendix A). TAGs eluted as packages, each with compounds of a comparable polarity (Figure 8), and this fact rendered a set of fragment peaks at a given time in the MS^E^ (see Figure 9 as an example), which were ascribed to each TAG according to the rules shown above (Appendix A).

## 3. Discussion

### 3.1. UPLC-MS Method

In this study, we aimed to determine whether the MS^E^ function could be used to assess TAG acyl chain composition in one chromatographic run. We first checked methanol and acetonitrile mixtures with 1% water and acidification with formic acid to render the [M + H]^+^ ions, but we obtained parent ions with poor intensity and partial fragmentation in the ESI source. This fact imposes a drawback to assessing acyl position. Indeed, TAG analysis through [M + H]^+^ ions has been shown to be better suited for an APCI source [7,13,26,27]. Consequently, taking into account previous work by other authors [11,12,26,28,29], we assayed different chromatographic methods for TAG separation, and we found that the method that better fitted our purpose was that in which non-aqueous solvents (i.e., acetonitrile and 2-propanol) with a reverse phase column were used. Additionally, following Byrdwell and Neff [7] and Leskinen et al. [26], we decided to use NH_4_OH instead of ammonium salts in order to avoid a high ion content in the elution non-aqueous solvents. These authors had already shown that a method of such characteristics could be suitable for TAG analysis and determination of fatty acyl position in the glycerol backbone.

Previously, Balgoma et al. [13], using the same instruments with methanol and 2-propanol as elution solvents, compared their fragmentation patterns of [M + NH_4_]^+^ ions in ESI with previous studies fragmenting [M + NH_4_]^+^ ions in ESI, and [M + H]^+^ ions in APCI. They found that the different adducts present the same fragmentation trends in different instruments. Nonetheless, they found that the degree of fragmentation was instrument and isomer dependent. Conversely, in the present study, equivalent fragmentation intensities from the [M + NH_4_]^+^ adduct were shown for all the TAGs, with linear equations being drawn for sn-1/sn-3 positions and for the sn-2 position. Nonetheless, these equations are yet to be refined, in addition to determining whether they can be used for diverse polyunsaturated fatty acyls and the position they esterify.

Even though fragment peaks show a rather low intensity (from 10^3^ to 10^4^) in the MS/MS (MS^E^) spectrum, it was enough to determine their elemental composition; additionally, this feature is useful in assessing what co-eluting TAG species correspond to, as shown above. Comparison of fragment intensity between methods reported in other studies is difficult, as generally, the concentration used is not provided or, otherwise, only the relative intensity is depicted in figures.

### 3.2. Ammonium Concentration Effect on Fragmentation

Currently, a large number of TAGs can be extracted from a biological sample, and chromatographic baseline separation of every species is almost impossible. Therefore, elucidation of the respective regioisomer structure becomes a pitfall. Obtaining well-resolved fragmentation spectra can be achieved to some extent by using multiple reaction monitoring methods (MRM) (that is, data-dependent acquisition (DDA)), but not all mass spectrometers offer such possibility. Additionally, a high number of transitions (reactions) in Q-ToF MS^2^ experiments may hamper proper spectrometer analyses because of overlapping dwell times. Furthermore, separate optimization for each transition is required [6].

In this study, we show that TAG regioisomer configuration can be elucidated in one chromatographic run with high accuracy using a MS^E^ method in the mass spectrometer. This strategy avoids the requirement of presetting MRM transitions. Similar to what was previously shown by other authors [12,13,17,30], release of the acyl chain at the sn-2 position was less favorable than the release of the acyl chains at the sn-1 and sn-3 positions. Indeed, the slope of the linear equations correlating the peak intensity of the diacylglycerol (DAG) carbocation resulting from the loss of the acyl chain at the sn-2 position was half (0.188) of that calculated for the sn-1 and sn-3 positions (0.369). Nonetheless, given that these constants were averaged for saturated and unsaturated acyls, slight variations may be observed in complex mixtures [13]. We have also shown that acyl chain loss is hampered by increased ammonium concentration, owing to the increased formation of the ammonium plus acetonitrile adduct ([M + ACN + NH_4_]^+^). This effect is likely due to the stabilization of a crystal-like network within the solution by hydrogen bonds and Van-der Waals forces between the carbonyls of the TAGs, the ammonium, and the resonant triple bond of the acetonitrile (CH_3_C≡N). Such a stabilized network may explain the lower acyl neutral loss with increased ammonium concentration. The same can be said for the formation of the sodium adduct. Furthermore, the formation of (M + ACN + NH_4_)_n_ clusters might reduce the mobility through the chromatographic column because of a higher size of the molecular entity and augmented interaction with the solid phase, which would elicit an increase in the retention times.

The currently accepted mechanistic explanation of acyl release through neutral loss points to an easier loss of the acyl chain at the sn-2 position because this chain may receive protons from the two adjacent acyl chains [6,31]. However, the opposite trend was observed with the ammonium adduct in previous studies and this study [12,17,30]. Hence, it may be hypothesized that the ammonium group gives rise to a resonant structure between the carbonyl of the sn-2 position and one of the carbonyls at the sn-1 and sn-3 positions in the [M + NH_4_]^+^ adduct [14]. This resonant structure would come free of one of the two acyl chains at the extreme position during time enough for the ester linkage being broken by the vibrational energy of the collision-induced dissociation mechanisms (CIDMs). This acyl freeing would randomly take place over different sn-1 or sn-3 position, which would explain the close similar probability (close constant) of losing both terminal acyl chains, even though the number of carbons and double bonds may influence the time duration it becomes free from the resonant complex generated by the ammonium ion. Conversely, but with a lower probability, combined movements of stretching of the two terminal methylenes of the glycerol moiety towards the center along with their rotation may have the acyl chain at the sn-2 position, away from the two terminal acyl chains and the ammonium, when vibrational energy of the CIDM is likely to break the ester bond of this acyl chain. Either loss of one acyl chain would result in carbocation of the diacylglycerol after the ammonium is also released. This mechanism might explain the differential release efficiency between sn-1 or sn-3 and the sn-2 acyl chains. Additional linkage with hydrogen bonds by the acetonitrile molecule would reduce the probability of any acyl chain being free of the resonant complex and, consequently, its release by the CIDM energy.

### 3.3. Ovine Plasma TAGs

The delayed elution and lower fragmentation efficiency with increased ammonium concentration were observed in the ovine plasma TAG analysis as for the standards (Figure 8 and Figure 9). Up to 53 species were identified with this method (Table 2). The longest acyl chain was 20 carbons, whilst the shortest was of 12 carbons. The highest number of unsaturations was 6, which included 4 from the eicosatetraenoic acid (TAG(20:4/18:1/18:1)). About 26 species accounted for C16 acyl chains, whereas 52 C18 acyl chains esterified 32 TAG species. Odd carbon acyls of C17 and C19 were found in 6 species.

When elution time was plotted against the number of double bonds for diverse TAGs (Figure 10), a quadratic equation was found to fit this correlation, whereas a power equation was found to better fit this correlation for the standards (Figure 3). These contradicting results may be a consequence of using only three points in the standard correlation, and consequently, a quadratic equation could not be fitted. There was agreement between the standards and the ovine plasma TAG species regarding fitting to an exponential equation for the correlation of elution time against the carbon number (Figure 11), with the coefficient of the exponent decreasing as the double bond number increases.

There are not many reports on ovine plasma triacylglycerols, but mainly on ovine milk and regarding diet influence on its composition [32,33,34,35,36]. Up to 134 molecular TAG species were reported by Fontecha et al. [33] in ovine milk, with about 84% having long-chain fatty acyls (C16 and C18), but no very long-chain fatty acyls were reported. In a comparative study between two ovine groups fed differently, some TAGs were shown to have a differential composition in milk from each group [36].

## 4. Materials and Methods

### 4.1. Reagents and Standards

Triglycerolipid standards were acquired from Larodan Research Grade Lipids (Solna, Sweden). Specifically, the following standards were purchased: 1,3-Heptadecanoin-2-Olein (reference 34-1728-7; TAG(52:1)), 1,2-Caprin-3-Stearin (reference 34-1000-7; TAG(38:0)), 1,3-Olein-2-Lignocerin (reference 34-1823-7; TAG(60:2)), Triheptadecanoin (reference 33-1700-7; TAG(51:0)), 1-Myristin-2-Olein-3-Behenin (reference 34-3014-7; TAG(54:1)), and 1,3-Olein-2-Myristin (reference 34-1830-7; TAG(50:2)). Methanol (code A456-212), Acetonitrile (code A955-212), and 2-Propanol (code A461-212) were LC/MS grade (OPTIMA^®^, Fisher Scientific™ S.L., Madrid, Spain). Ethanol (HPLC grade) was from Scharlau (Multisolvent^®^, reference ET00152500), and Acetone (reference 1.00020.2500) was from Merck KGaA (Darmstadt, Germany). All experiments were conducted at a concentration of 10 μg/mL, which rendered mM concentrations (×10^−2^) of 1.44 for TAG(38:0), 1.20 for TAG(50:2), 1.18 for TAG(51:0), 1.16 for TAG(52:1), and 1.13 for TAG(54:1).

### 4.2. Liquid Chromatography (UPLC)

Lipid extraction was carried out by the chloroform–methanol method [37], but dichloromethane:methanol (2:1 *v*/*v*) was used instead. After the organic solvent was evaporated to dryness in a speed-vac (model DNA-23050-B00, MiVac, Genevac), the remaining pellet was dissolved in a mixture of ethanol–acetone–2-propanol (1:1:1 *v*/*v*/*v*). TAG standards were also dissolved in the same solvent mixture. A volume of 7.5 μL of this solution was injected into the liquid chromatographer. All samples were analyzed in triplicate.

Ultraperformance liquid chromatography (UPLC) was conducted in an Acquity™ UPLC system (WATERS, Manchester, UK) equipped with an automatic injector (Sample Manager) and a binary solvent pump (Binary Solvent Manager). The chromatographic column was an Acquity UPLC BEH HSS T3 100 × 2.1 mm, 1.7 μm p.s., with a 10 × 2.1 mm precolumn (vanguard column) of the same stationary phase. Solvents were (A) acetonitrile–2-propanol–methanol (3:4:3 *v*/*v*/*v*) and (B) acetonitrile–2-propanol (3:7 *v*/*v*), both with 0.1–0.6% of 32% aqueous NH_4_OH. Compounds were eluted at a flow rate of 0.40 mL/min using the following gradients: initial, 100% A; 3 min, 100% A; 6 min, 98% A; 8 min, 98% A; 9.5, min 95% A; 11 min, 95% A; 16 min, 100% A; and 18 min, 100% A. In order to avoid the carry over, which was calculated to be about 12% for saturated TAGs and about 5% for unsaturated TAGs, methanol was injected and an entire chromatographic run was performed after each sample analysis, with an additional one after five samples were injected.

### 4.3. Mass Spectrometry

After the chromatographic separation, TAG species were detected by mass spectrometry with positive ionization using an MS^E^ method [38]. The column output was connected to an electrospray ionization source (ESI, Z-spray) of a mass spectrometer SYNAPT G2 HDMS (Waters, Manchester, UK), with time-of-flight analyzer (QToF). Acquisition parameters were set as follows: capillary, 0.8 kV; sampling cone, 15 V; source temperature, 90 °C; desolvation temperature, 280 °C; cone gas, 40 L/h; and desolvation gas, 700 L/h.

Data were acquired with the software MassLynx™ (Waters, Manchester, UK) at a rate of 5 scans/s within the range 0–18 min, with *m*/*z* 100–1200 Da for the low-energy function and *m*/*z* 100–900 Da for the high-energy function (MS^E^ method, trap collision energy of 30 V). LC and MS methods were optimized using the commercial standards shown above. Peak areas and heights were measured using the QuanLynx^®^ software (Waters, Manchester, UK).

## 5. Conclusions

In this work, a UPLC-MS^E^ method is presented for triacylglycerol analysis where the dehydroxydiacylglycerol fragments can be determined accurately. Increasing ammonium hydroxide concentrations in the eluent led to increased formation of the [M + AcN + NH_4_]^+^ adduct, whilst the [M + NH_4_]^+^ adduct dropped beyond a concentration of about 18.64 mM. Stabilization with acetonitrile of the ammonium adduct complex dwindled the fragmentation intensity. This behavior suggests the formation of a resonant complex between the ammonium and the carbonyls of the fatty acyls that becomes fragmented, yielding the neutral loss of the acyl chains. Fragmentation of the fatty acyl at the sn-2 position is less favored than that of the fatty acyls at the sn-1 and sn-3 positions. Fragment intensity from either position can be calculated from linear equations that relate the fragment intensity to the intensity of the [M + NH_4_]^+^ adduct in the full scan. Further factors involved in TAG analysis by UPLC-MS still require research in order to improve regioisomer identification and TAG quantification. The results shown here are expected to contribute to progress in triacylglycerol analysis. Further work will aim at stablishing the optimal conditions for both regioisomer structure identification and quantification in one run in biological samples. A software freely available for automation of acyl chain ascription may be developed taking into consideration the results of this study.

## Figures and Tables

**Figure 1 metabolites-12-00452-f001:**
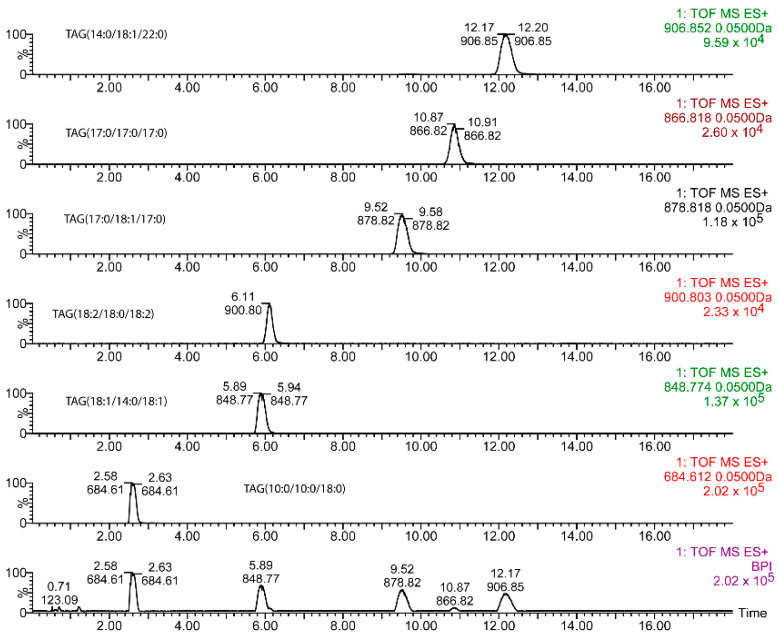
Typical base peak chromatogram (lower panel) and extracted ion chromatograms for the diverse triacylglycerols (TAGs) used in this study. TAG concentration was 10 μg/mL (see Section 4 for mM concentration equivalence).

**Figure 2 metabolites-12-00452-f002:**
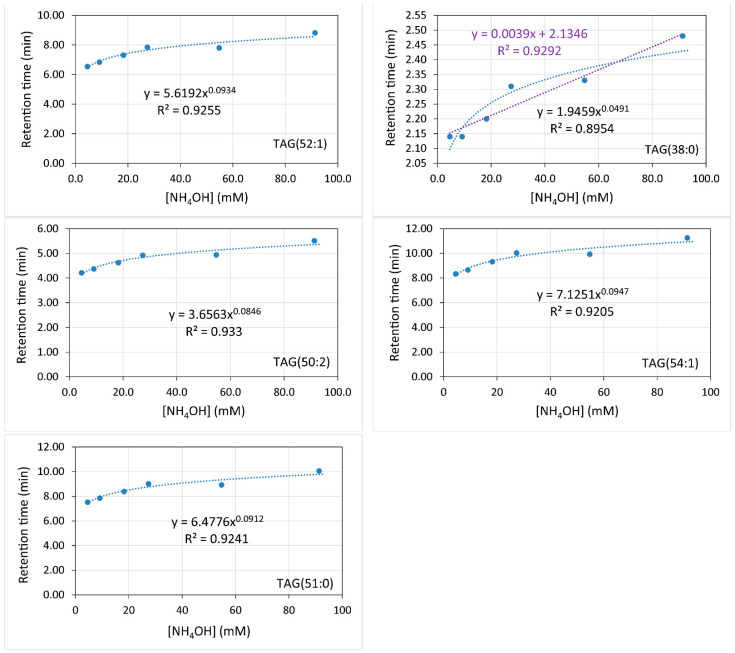
Variation of elution time with NH_4_OH concentration for diverse triacylglycerol standards.

**Figure 3 metabolites-12-00452-f003:**
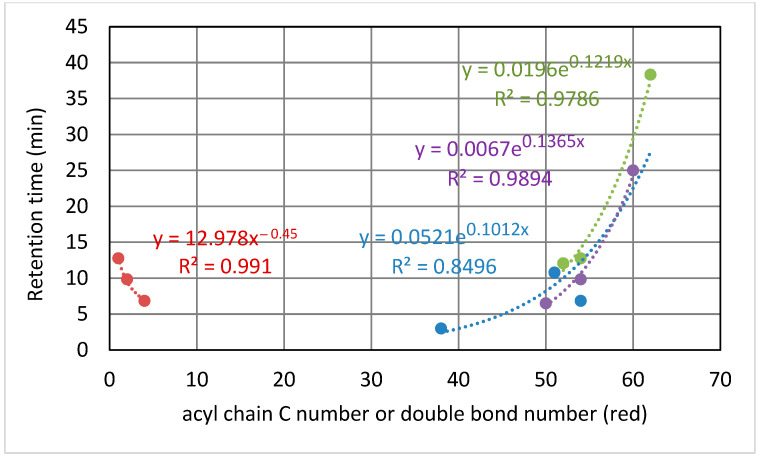
Elution time with the carbon number of the acyl chains for the same number of unsaturations (green, one double bond; violet, two double bonds; and blue, merged C number irrespective of the double bonds) and with the double bond number for 54 acyl carbons (red). Data for [NH_4_OH] = 9.13 mM.

**Figure 4 metabolites-12-00452-f004:**
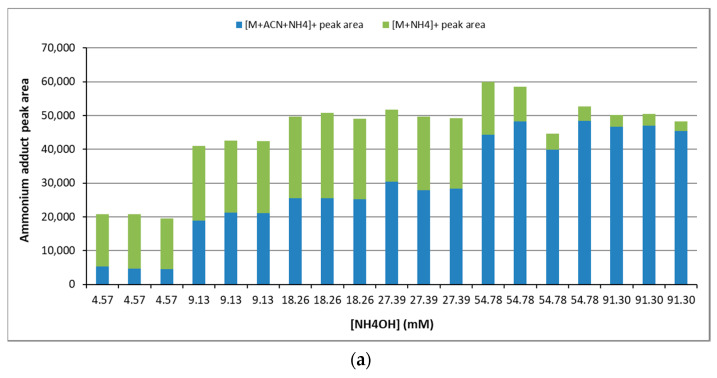
Chromatographic peak area of the adducts formed at different concentrations of NH_4_OH (**a**) for TAG(17:0/18:1/17:0) and (**b**) for TAG(14:0/18:1/22:0). Green triangles, [M + NH_4_]^+^ adduct; blue diamonds, [M + ACN + NH_4_]^+^ adduct; red triangles, sum of the two adducts; blue line, polynomial trend of [M + ACN + NH_4_]^+^ adduct formation (see Appendix A for further details). ACN means acetonitrile.

**Figure 5 metabolites-12-00452-f005:**
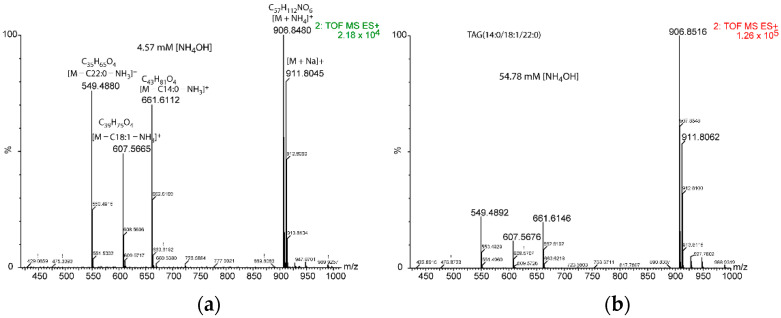
Comparative fragmentation intensity under two ammonium concentrations for TAG(14:0/18:1/22:0). Adducts and elemental compositions are only shown in panel (**a**) for the sake of clarity. Relative intensity of the [M + NH_4_]^+^ adduct was 2.18 × 10^4^ for 4.57 mM [NH_4_OH] (panel (**a**)) and 1.26 × 10^5^ for 54.78 mM [NH_4_OH] (panel (**b**)).

**Figure 6 metabolites-12-00452-f006:**
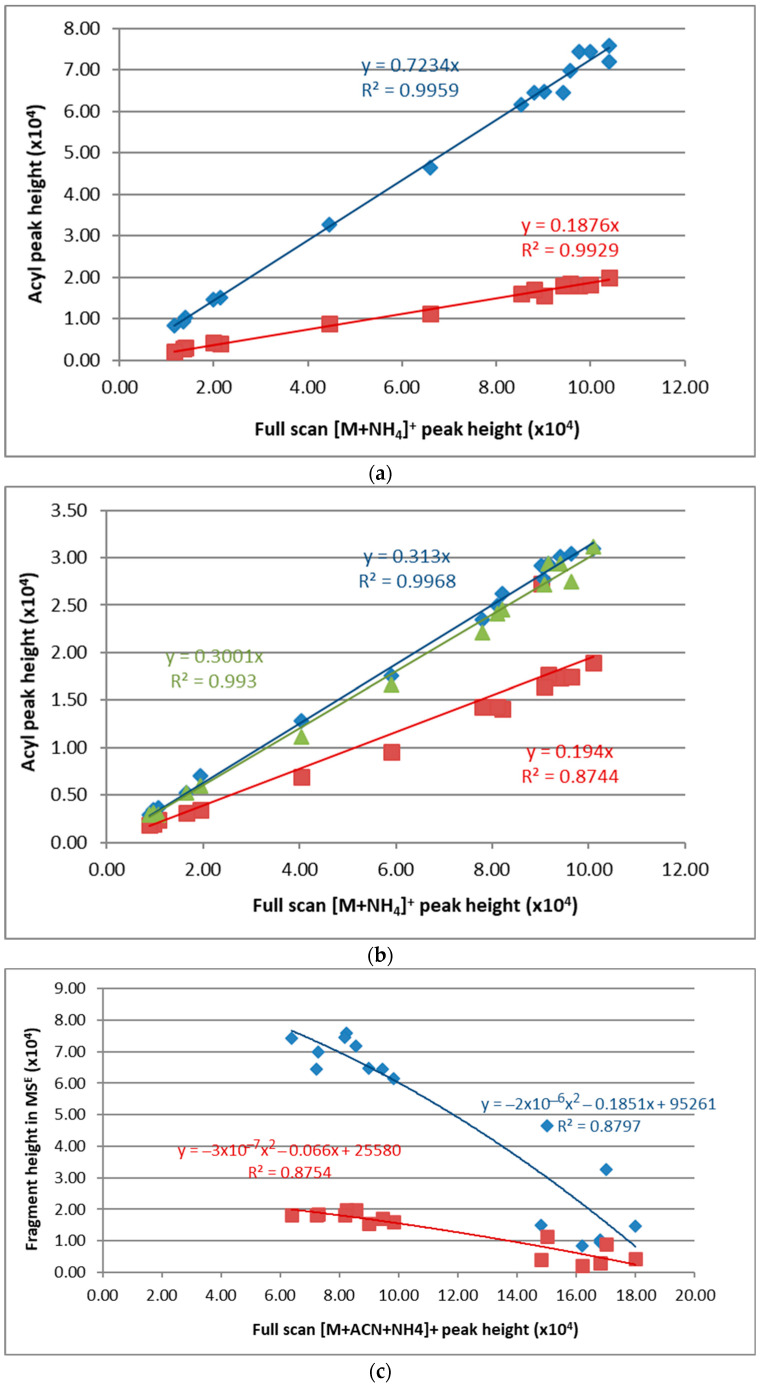
Correlation of the deshydroxydiacylglycerol (acyl in the Y-axes) peak heights in the MS^E^ spectrum with the [M + NH_4_]^+^ (**a**,**b**) or the [M + ACN + NH_4_]^+^ (**c**,**d**) peak heights in the full scan. The peak intensities varied with the [NH_4_OH]. Results are shown for two triacylglycerols: (**a**,**c**) for TAG(17:0/18:1/17:0); blue diamonds, acyl in *sn*-1 and *sn*-3 positions (C17:0); red squares, acyl in *sn*-2 position (C18:1); (**b**,**d**) for TAG(14:0/18:1/22:0); blue diamonds, acyl in *sn*-1 position (C22:0); green triangles, acyl in *sn*-2 position (C18:1); red squares, acyl in *sn*-3 position (C14:0) Adscription of *sn*-1 and *sn*-3 positions is arbitrary. See Appendix A for other standards.

**Figure 7 metabolites-12-00452-f007:**
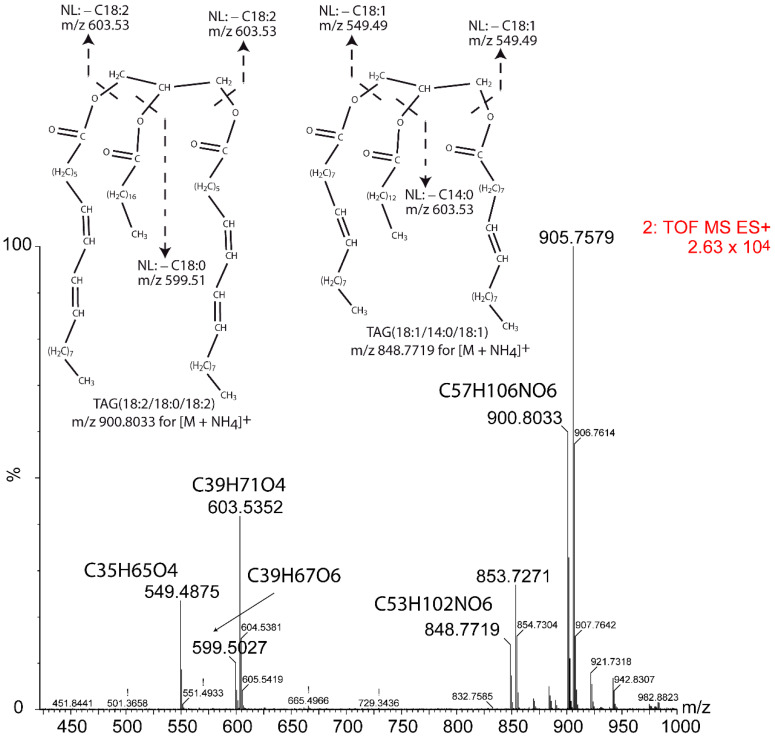
MS^E^ mass spectrum at 6.29 min of two partially co-eluting TAGs. TAG(54:4) has *m*/*z* 900.8033 and elemental composition C_57_H_106_NO_6_ for the [M + NH_4_]^+^ adduct; TAG(50:2) has *m*/*z* 848.7719 and elemental composition C_53_H_102_NO_6_ for the [M + NH_4_]^+^ adduct. The elemental composition of each fragment is also indicated. NL means neutral loss.

**Figure 8 metabolites-12-00452-f008:**
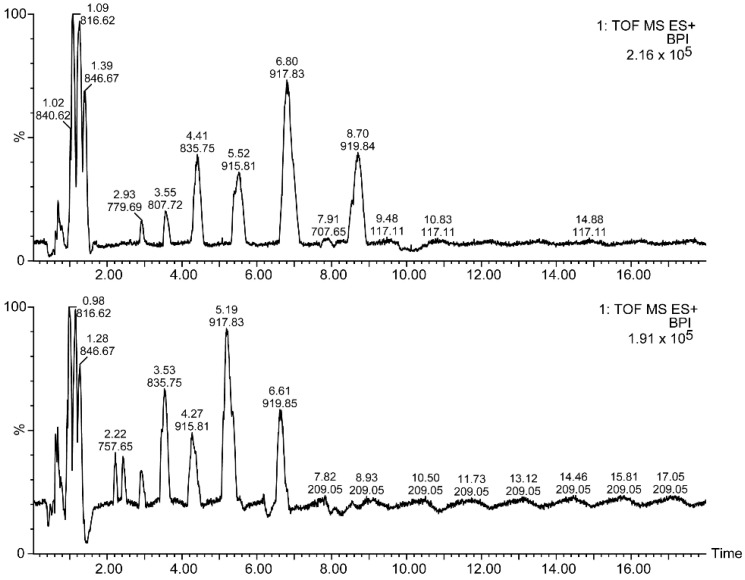
Base peak chromatograms of TAGs from ovine plasma at two NH_4_OH concentrations in the eluents: 54.78 mM (**upper** panel) and 9.13 mM (**lower** panel). The elution time and the *m*/*z* of the most intense peak is shown over each chromatographic peak.

**Figure 9 metabolites-12-00452-f009:**
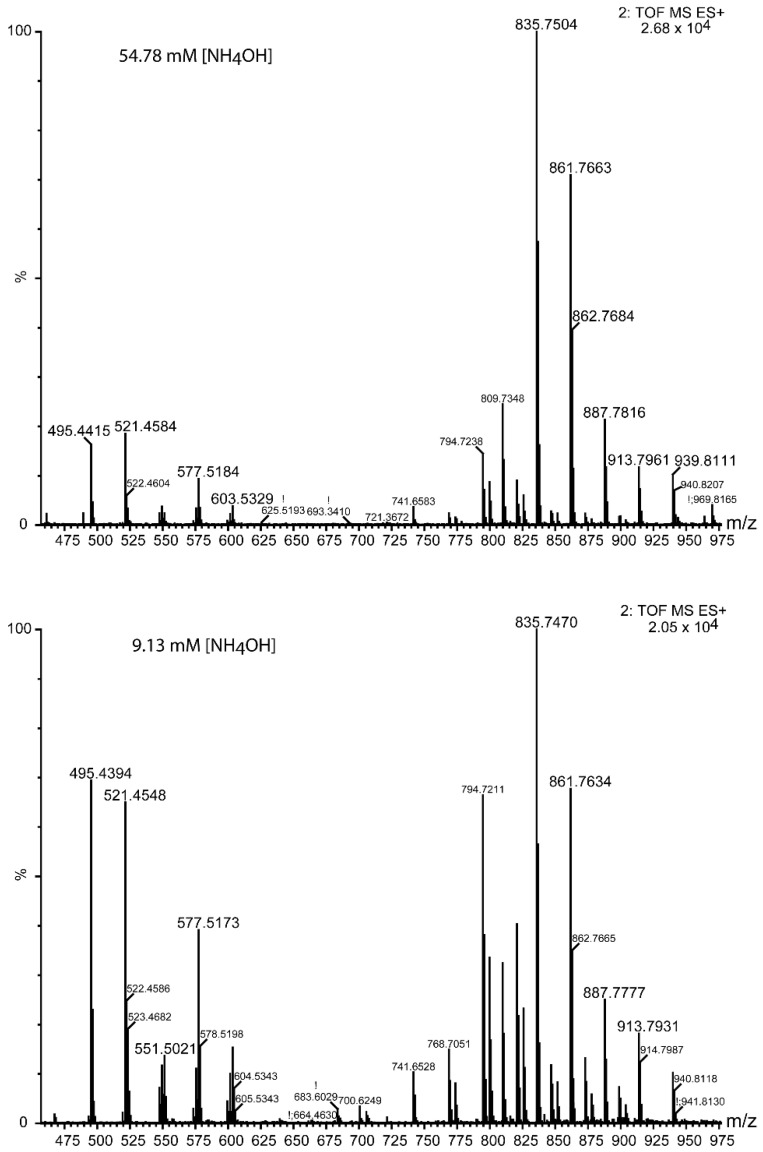
Example of a set of fragment peaks at a given time in the MS^E^.

**Figure 10 metabolites-12-00452-f010:**
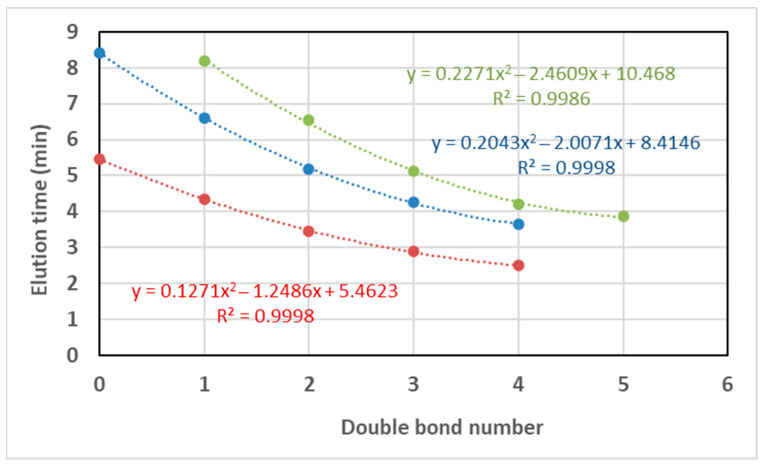
Correlations of elution time (min) against the double bond number for TAGs with 48 (red), 52 (blue), and 54 (green) carbons for 9.13 mM [NH_4_OH].

**Figure 11 metabolites-12-00452-f011:**
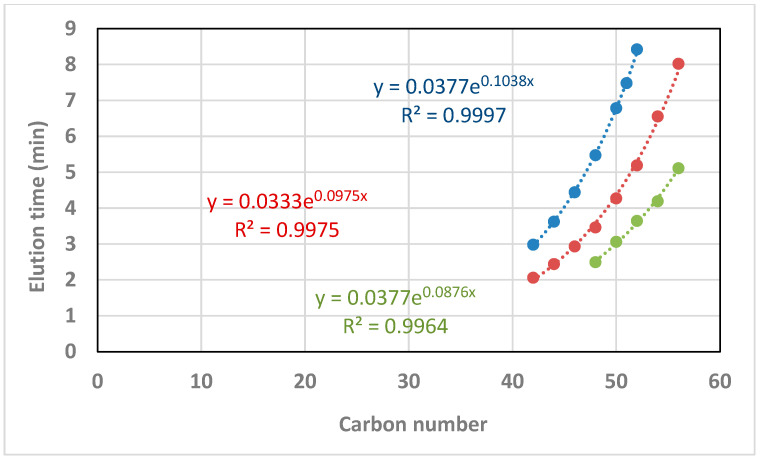
Correlations of elution time (min) against the carbon number for TAGs with 0 double bonds (blue), 2 double bonds (red), and 4 double bonds (green) for 9.13 mM [NH_4_OH].

**Table 1 metabolites-12-00452-t001:** Example of fragment ascription for composition determination of two partially co-eluting TAGs.

Fragment *m*/*z*	Fragment Elemental Composition	Retention Time (min)	MS^E^ Peak Intensity	TAG	TAG Elemental Composition	TAG MS Peak Intensity	Fatty Acyl Lost	Potential TAG	Calculated Fragment Intensity	
									*sn*-1 or *sn*-3 position	*sn*-2 position
				(54:4)						
549.49	C_35_H_65_O_4_	5.89	6190		C_57_H_102_O_6_	15,800	C22:3	22:3/-/-	5830.2	2970.4
599.51	C_39_H_67_O_4_	6.12	2670		C_57_H_102_O_6_	15,800	C18:0	18:0/18:2/18:2	5830.2	2970.4
603.53	C_39_H_71_O_4_	5.94	11,000		C_57_H_102_O_6_	15,800	C18:2	18:2/18:0/18:2	5830.2	2970.4
				(50:2)						
549.49	C_35_H_65_O_4_	5.89	6190		C_53_H_98_O_6_	3700	C18:1	18:1/14:0/18:1	1365.3	695.6
599.51	C_39_H_67_O_4_	6.12	2670		C_53_H_98_O_6_	3700	C18:2	18:2/14:0/-	1365.3	695.6
603.53	C_39_H_71_O_4_	5.94	11,000		C_53_H_98_O_6_	3700	C14:0	14:0/18:1/18:1	1365.3	695.6

**Table 2 metabolites-12-00452-t002:** Triacylglycerols (TAG) identified in an ovine plasma sample. More than one structure is proposed where no resolution was feasible. The regioisomery was established according to the rules found for the standard compounds. Rt means retention time.

*m*/*z* [M+ACN+NH_4_]^+^	*m*/*z* [M+NH4]^+^	Rt (min) 9.13 mM [NH_4_OH]	Elemental Composition [M+ACN+NH_4_]^+^	TAG	Acyl Chains
697.6086	656.5835	1.76	C41H81N2O6	TAG(36:0)	(12:0/12:0/12:0)
723.6227	682.589	1.76	C43H83N2O6	TAG(38:1)	(12:0/12:0/14:1)
749.6391	708.6163	1.78	C45H85N2O6	TAG(40:2)	(12:0/14:1/14:1)
775.6596	734.6071	1.78	C47H87N2O6	TAG(42:3)	(14:1/14:1/14:1) Too low
823.7106	782.6843	1.72	C49H95N2O7	Oxidized TAG?	TAG(44:1)(OH)
849.7297		1.76	C51H97N2O7	Oxidized TAG?	TAG(46:2)(OH)
877.7574		1.95	C53H101N2O7	Oxidized TAG?	TAG(48:2)(OH)
725.6398	684.6135	2.09	C43H85N2O6	TAG(38:1)	(12:0/12:0/14:0)
751.6572	710.6262	2.06	C45H87N2O6	TAG(40:1)	(12:0/12:0/16:1)
777.6715	736.6446	2.06	C47H89N2O6	TAG(42:2)	(14:1/16:1/12:0) (14:1/14:1/14:0)
803.688	762.6505	2.11	C49H91N2O6	TAG(44:3)	(14:1/14:1/16:1)
753.6713	712.643	2.48	C45H89N2O6	TAG(40:0)	(12:0/12:0/16:0)
779.6881	712.6439	2.42	C47H91N2O6	TAG(42:1)	(14:1/16:1/12:0)
805.7037	764.6792	2.44	C49H93N2O6	TAG(44:2)	(14:1/16:1/14:0) (14:1/16:0/14:1)
831.7203	790.6948	2.48	C51H95N2O6	TAG(46:3)	(18:1/12:0/16:2)
857.7362	816.7054	2.49	C53H97N2O6	TAG(48:4)	(18:2/18:2/12:0)
781.702	740.6749	2.98	C47H93N2O6	TAG(42:0)	(16:0/14:0/12:0) (18:0/12:0/12:0) (14:0/14:0/14:0)
807.7178	766.6906	2.91	C49H95N2O6	TAG(44:1)	16:1/14:0/14:0
833.7346	792.7086	2.93	C51H97N2O6	TAG(46:2)	(16:1/12:0/18:1)
859.7489	818.7203	2.89	C53H99N2O6	TAG(48:3)	(18:1/12:0/18:2)
885.7668	844.7361	3.06/3.25	C55H101N2O6	TAG(50:4)	(14:1/16:1/20:2) (16:2/14:0/20:2)
809.7333	768.7072	3.62	C49H97N2O6	TAG(44:0)	(16:0/14:0/14:0)
835.7502	794.7231	3.53	C51H99N2O6	TAG(46:1)	(18:1/16:0/12:0)
861.7655	820.7371	3.46	C53H101N2O6	TAG(48:2)	(18:1/12:0/18:1)
887.7794	846.7521	3.50	C55H103N2O6	TAG(50:3)	(18:1/16:1/16:1)
913.7956	872.7681	3.64	C57H105N2O6	TAG(52:4)	(16:1/18:1/18:2)
939.8111	898.7817	3.88	C59H107N2O6	TAG(54:5)	(18:2/18:2/18:1)
965.8273	924.7947	3.84	C61H109N2O6	TAG(56:6)	(20:4/18:1/18:1)
837.7642	796.7369	4.44	C51H101N2O6	TAG(46:0)	(16:0/14:0/16:0)
863.7798	822.7529	4.33	C53H103N2O6	TAG(48:1)	(16:0/18:1/14:0) (18:1/16:0/14:0)
889.7956	848.7697	4.27	C55H105N2O6	TAG(50:2)	(18:1/16:0/16:1)
915.8111	874.7841	4.27	C57H107N2O6	TAG(52:3)	(18:1/16:1/18:1)
941.8257	900.7987	4.19	C59H109N2O6	TAG(54:4)	(18:1/18:2/18:1)
967.8471	926.8208	4.56	C61H111N2O6	TAG(56:5)	(20:2/18:2/18:1)
993.8668	952.7939	4.53	C63H113N2O6	TAG(58:4)	(20:2/20:1/18:1)
891.8107	850.7844	5.35	C55H107N2O6	TAG(50:1)	(18:1/16:0/16:0)
917.8276	876.8007	5.19	C57H109N2O6	TAG(52:2)	(18:1/18:1/16:0)
943.8428	902.8156	5.11	C59H111N2O6	TAG(54:3)	(18:1/18:1/18:1)
969.8508	928.8232	5.11	C61H113N2O6	TAG(56:4)	(20:2/18:1/18:1)
865.7928	824.7779	5.47	C53H105N2O6	TAG(48:0)	(18:0/16:0/14:0) (14:0/20:0/14:0)
905.8237	864.8032	5.93	C56H109N2O6	TAG(51:1)	(18:1/17:0/16:0)
931.8417	890.8013	5.84	C58H111N2O6	TAG(53:2)	(18:1/17:0/18:1)
957.8604	916.8351	5.73	C60H113N2O6	TAG(55:3)	(19:1/17:1/19:1)
893.8301	878.8172	6.78	C55H109N2O6	TAG(50:0)	(16:0/18:0/16:0)
919.8469	878.8172	6.61	C57H111N2O6	TAG(52:1)	(18:1/18:0/16:0)
945.8621	904.8362	6.55	C59H113N2O6	TAG(54:2)	(18:1/18:0/18:1)
971.8732	930.8461	6.49	C61H109N2O6	TAG(56:3)	(20:1/18:1/18:1)
907.8419	866.8082	7.48	C56H111N2O6	TAG(51:0)	(17:0/17:0/17:0)
933.8631	892.8331	7.38	C58H113N2O6	TAG(53:1)	(18:1/17:0/18:0)
959.8741	918.8602	7.25	C60H115N2O6	TAG(55:2)	(18:1/18:0/19:1)
973.8895	932.8611	8.02	C61H117N2O6	TAG(56:2)	(18:1/18:0/20:1)
947.8745	906.8404	8.19	C59H115N2O6	TAG(54:1)	(18:1/18:0/18:0) (20:1/16:0/18:0)
921.8611	880.8317	8.42	C57H113N2O6	TAG(52:0)	(18:0/18:0/16:0)

## Data Availability

The data presented in this study are available in the Appendix A.

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
