# Peer review of "Ammonia Concentration in the Eluent Influences Fragmentation Pattern of Triacylglycerols in Mass Spectrometry Analysis"

_metabolites, 2022, doi:10.3390/metabo12050452_

Round 1

Reviewer 1 Report

The review of Velasco et al. provides the LC-MS method for analysis of triacylglycerols (TAG). However, minor revisions are required in the manuscript to provide explanation about the presented method’s originality in comparison to previously published manuscripts.

The classic method for TAG sample preparation using ESI-MS would be the acidification of samples with formic acid (FA) or other weak acids. It is not clear why the author decided to use NH4OH.  The authors should further discuss their reasoning for using NH4OH instead of weak acids.

The concentration that is used in Figure 1 is 10 mM of TAG. This concentration is very high, probably due to the preparation of samples in basic condition. The intensity of the peak height is also low (in some cases below to 1e5). The author should discuss the limited concentration of TAG that can be used by the developed method.

In Figure 4, that describe “Chromatographic peak area of the adducts formed at different concentrations of NH4OH”, there are important differences for [M+ACN+NH4]+  at different concentrations of NH4OH. The R2 value also shows that the quadratic equation does not fit well. The author should discuss more about these variations.

Did the authors check the ratio of [M+ACN+NH4]+   to  [M+NH4]+   if source temperature increased > 100 C?

Figure 7 should contain the structure of TAG and indicate the position that the molecule fragmented. It is difficult to follow the conclusion of the author without the structure of TAG.

It is unclear what the concentration dependence of TAG on the area of the precursor ion and whether the developed method can be used for quantitative evaluation of TAG from biological samples.

It is difficult to evaluate the originality of the method developed to analyze TAG in comparison to other studies. The author should further discuss the originality and compare it to other studies.

Author Response

The authors thank reviewer's comments.

Rev: The review of Velasco et al. provides the LC-MS method for analysis of triacylglycerols (TAG). However, minor revisions are required in the manuscript to provide explanation about the presented method’s originality in comparison to previously published manuscripts.

The classic method for TAG sample preparation using ESI-MS would be the acidification of samples with formic acid (FA) or other weak acids. It is not clear why the author decided to use NH4OH.  The authors should further discuss their reasoning for using NH4OH instead of weak acids.

Answer: The reviewer’s comment is relevant and we acknowledge this deficiency in our manuscript. Nonetheless, we should say there are enough references in the bibliography stating that acidification to render the [M+H]+ adduct is better suited for APCI than for ESI, in which the [M+cation]+ adduct gives improved results (Leskinen et al., 2007). Additionally, the same authors (Leskinen et al., 2007), and others previously (see Byrdwell and Emken, 2002), showed that NH4OH could be used instead of ammonium salts with acetonitrile and 2-propanol to render [M+NH4]+ adducts with high intensity peaks. This explanation in now added to the Discussion section as subsection 3.1.   

Rev: The concentration that is used in Figure 1 is 10 mM of TAG. This concentration is very high, probably due to the preparation of samples in basic condition. The intensity of the peak height is also low (in some cases below to 1e5). The author should discuss the limited concentration of TAG that can be used by the developed method.

Answer: Thanks for reviewer’s comment. This concentration is a mistake. The actual concentration was 10 mg/mL, which corresponds to a slightly different mM concentration for every TAG species, ranging from 1.13 10-2 to 1.44 10-2 mM. This is now indicated in the Figure 1 and in the Materials & Methods section. We earnestly apologize for this mistake. In this figure it can be seen that only two TAGs have intensities below 105 and even one of them has an intensity of 9.54 104. Lower intensities are common in the fragmentation spectra.

Rev: In Figure 4, that describe “Chromatographic peak area of the adducts formed at different concentrations of NH4OH”, there are important differences for [M+ACN+NH4]+  at different concentrations of NH4OH. The R2 value also shows that the quadratic equation does not fit well. The author should discuss more about these variations.

Answer: The reviewer is right and, indeed, a logarithmic equation fits a bit better for most of the TAGs. Nonetheless, it should be said that the objective of this figure was only to show that the ionization efficiency increases with the ammonium concentration, even though at expenses of concomitantly rising the [M+ACN+NH4]+ adduct and lowering the fragmentation efficiency. Therefore, since fitting adduct formation trend to an equation does not actually have much relevance at present, we have decided to change the figure format to bar graph and remove equation fitting in both the main text and supplementary information file. Further experiments will aim at narrowing the ammonium concentrations that may serve to find the optimal one after fitting them to a more reliable equation - a Gompertz equation (a*exp(b*exp(-log(conc))) might apparently be appropriate for it -.

Rev: Did the authors check the ratio of [M+ACN+NH4]+   to  [M+NH4]+   if source temperature increased > 100 C?

Answer: Thanks for this comment. Actually, we checked 3 source temperatures, namely 90, 100 and 120 C and we found the better performance happened with 100 C. Nonetheless, results presented in this study come from a first approach, and further experiments will aim at thoroughly testing combination of different parameters, which include column, desolvation and source temperature, CID energy, and solvent proportion in order to find the most convenient trade-off between ionization and fragmentation efficiencies for proper species identification and quantification.   

Rev: Figure 7 should contain the structure of TAG and indicate the position that the molecule fragmented. It is difficult to follow the conclusion of the author without the structure of TAG.

Answer: The figure has been modified to add the TAG structures with neutral losses (NL) rendering the diverse fragments being illustrated. We hope this modification helps better understanding the explanation provided in the text. A correction has also been done in the figure legend as TAG(50:4) is actually TAG(50:2). The proper corrections have also been done in the text and in Table 1.

Rev: It is unclear what the concentration dependence of TAG on the area of the precursor ion and whether the developed method can be used for quantitative evaluation of TAG from biological samples.

Answer: The reviewer is right. Quantification is a relevant issue. In this study, however, we have focused on structure elucidation and how the ammonium concentration influences ionization and fragmentation efficiency. Further work will aim at stablishing the optimal conditions for both structure elucidation and quantification. Nonetheless, we can say that previous experiments have shown that a measurable signal (between 4.5 103 and 1.1 104) is obtained with 54.78 mM [NH4OH] for the [M+ACN+NH4]+ adduct, but weak signal is rendered for the [M+NH4]+ adduct, at a TAG concentration of 2 mg/mL. As indicated above, other factors need to be already improved.

Rev: It is difficult to evaluate the originality of the method developed to analyze TAG in comparison to other studies. The author should further discuss the originality and compare it to other studies.

Anser: Our method aimed at using the MSE function to analyse TAGs in one run for the first time. It is shown that the structure of even co-eluting TAGs can be characterized with this method. It is explained in detail how the fragment intensities in the MSE spectrum in relation to the intensity of the parent ion in the full scan can be used to fatty acyl ascription to the co-eluting TAGs. Additionally, equations are provided to calculate the expected intensity of any fragment according to the sn position it esterifies. Abundant data are also provided regarding how ionization and fragmentation are affected by the hydroxide ammonium in the eluents. These facts are stated in the discussion and in the conclusion sections.  

Reviewer 2 Report

The authors describe an article entitled “Ammonia concentration in the eluent influences fragmentation pattern of triacylglycerols in mass spectrometry analysis”. The topic of the manuscript is interesting, and the manuscript constitutes an interesting research article concerning the determination of experimental conditions for mass spectroscopy analysis.

The work is well-written and a well-constructed introduction has been established by the authors. Sufficient spectra and figures are included in the manuscript for comprehension and clarity. Interesting and convincing results are also presented in this work. Overall, I think that this is a manuscript that I recommend for publication after inclusion of minor revisions.

1) Authors demonstrated the impressive results concerning mass analyses of triacylglycerols. Can this method be generalized to other derivatives ?

2) In the conclusion section, authors summarized their works. Besides, no perspectives are provided. Please modify.

3) What is the pH of the solution analyzed ?

4) Quality of Figure 2 should be improved. At present, it’s difficult to read.

.

Author Response

Reviewer #2

The authors describe an article entitled “Ammonia concentration in the eluent influences fragmentation pattern of triacylglycerols in mass spectrometry analysis”. The topic of the manuscript is interesting, and the manuscript constitutes an interesting research article concerning the determination of experimental conditions for mass spectroscopy analysis.

The work is well-written and a well-constructed introduction has been established by the authors. Sufficient spectra and figures are included in the manuscript for comprehension and clarity. Interesting and convincing results are also presented in this work. Overall, I think that this is a manuscript that I recommend for publication after inclusion of minor revisions.

Authors: We thank the reviewer’s positive evaluation of our study and his/her valuable comments.

Rev: 1) Authors demonstrated the impressive results concerning mass analyses of triacylglycerols. Can this method be generalized to other derivatives ?

Answer: Actually, we are not sure to what derivatives the reviewer is referring. Nonetheless, we want to comment that the method is planned to be applied to diacylglycerols, which have less complex structure. And, of course, the method will be extended to oxidized triacylglycerols. We do not have thought of molecules with different molecular structures at the present.   

Rev: 2) In the conclusion section, authors summarized their works. Besides, no perspectives are provided. Please modify.

Answer: Done (see at the end of the conclusion section in blue).

Rev: 3) What is the pH of the solution analyzed ?

Answer: As it could be expected, pH is highly basic, with values ranging from 10.0 to 11.8 depending on the ammonium concentration and the solvent A or B. Regarding the TAG solution that is analyzed, we do not have micro-electrodes to accomplish such measurements in a small volume of 120 mL. Additionally, we should say that we consider that in non-aqueous solutions, even with no protic solvents as acetonitrile, the pH value is poorly reliable. We expect the reviewer agrees with our consideration.   

Rev: 4) Quality of Figure 2 should be improved. At present, it’s difficult to read.

Answer: The figure quality has been improved
